# Application of nnU-Net in kidney segmentation

Elaine Liu[1]

`347163903@qq.com`

**Abstract.** NnU-Net is a excellent deep learning framework that achieves state-of-the-art segmentation of medical images in some international competition tasks. In this paper, we train nnU-Net to the 2021 Kidney and Kidney Tumor Segmentation Challenge and attempt to mix the kidney data collected by ourselves.

**Keywords:** NnU-Net, segmentation, medical images

## 1 Introduction

In recent years, the incidence rate of renal tumors is increasing in adults. It has become one of the most common malignant tumors. Fortunately, early surgery can effectively treat the above diseases, and the semantic segmentation of kidney and tumor tissue is an important part of many clinical applications in urology, including artificial intelligence algorithm assisted doctor diagnosis system, surgical scheme planning, intraoperative assistance or tumor growth monitoring.

In the medical field, a large number of research papers have proposed architecture changes and extensions to improve performance. These studies are difficult for non experts to understand and even for experts to evaluate. About 12000 studies cited the 2015 U-net architecture for biomedical image segmentation, many of which proposed expansion and progress. Therefore, the author of nnU-Net puts forward such a hypothesis: if the appropriate processing process can be designed, the basic U-net architecture is difficult to be defeated[2].

This paper summarizes and study the successful experience of nnU-Net, and adds our own data to train the nnU-Net net model to segment kidney and kidney tumors as our submission.

## 2 Methods

Nnunet can make full use of the characteristics of the data set to train the basic u-net model without manual intervention, so as to surpass various modified u-net architectures.

### 2.1 Training and Validation Data

Our submission made use of the official KiTS21 training set and our own dataset. Our data came from kidney CT of 134 patients in the Tianjin Medical University General Hospital and the Second Xiangya Hospital Central South University.

### 2.2 Preprocessing

In the process of medical image processing, it is very important to keep the spacing of the image consistent and the specific size in the resampling process: one of the important motivations proposed by conv operation in CNN is that similar blocks in the image can extract features with shared convolution. Therefore, resampling all images can reduce the inconsistency between different images, It is convenient for convolution operation to extract common features. In

this article, we first try to use the value of the first place in the kits19 challenge. We resampled all cases to $3.22 \times 1.62 \times 1.62$ mm common voxel spacing.[3]

Hu (hounsfiled unit) value is an inevitable thing in medical image data preprocessing, which reflects the X-ray absorption degree of the organization. Take the absorption degree of water as a reference, that is, Hu = 0, the attenuation coefficient greater than water is straight, and the attenuation coefficient less than water is negative. Taking the Hu value of bone cortex and air as the upper and lower limits is an inevitable thing in medical image processing technology.

$$HU = pixel\_val * slope + intercept$$

If slope is 1 and intercept is 0, conversion is not required. If slope is not 1 and intercept is not 0, conversion is required.

### 2.3 Proposed Method

The author of nnU-Net makes this technology successfully used in 3D biomedical image segmentation. nnU-Net can automatically adapt to any data set. It realizes out of the box segmentation due to the following two key factors:
1. The pipeline optimization problem is formulated according to the data fingerprint (representing the key attributes of the data set) and the pipeline fingerprint (representing the key design selection of the segmentation algorithm).[2]
2. By condensing domain knowledge into a set of heuristic rules to clarify their relationship, the rules will stably generate high-quality pipeline fingerprints from the corresponding data fingerprints considering the associated hardware constraints. [2]

NnU-Net defines dataset fingerprint and pipeline fingerprint. Dataset fingerprint is the key representation of dataset, such as image size, voxel spatial information and category proportion; Pipeline fingerprint is divided into three groups: blueprint, interpolated and empirical parameters. Blueprint represents basic architecture design, such as u-net class template, loss function, training strategy and data enhancement; Inferd represents encoding the necessary adaptation of the new dataset and includes modifications to the exact network topology, patch size, batch size and image preprocessing. The relationship between data fingerprint and inford is established by executing a set of heuristic rules. When applied to invisible data sets, there is no need for expensive re optimization. Through the cross validation of training cases, the empirical parameters can be determined automatically. By default, nnU-Net generates three different u-net configurations: a 2D u-net, a 3D u-net running at full image resolution, and a 3D u-net cascade. After cross validation, nnU-Net will select the best configuration or overall performance according to experience. Dataset fingerprint and pipeline fingerprint are defined. Dataset fingerprint is the key representation of dataset, such as image size, voxel spatial information and category proportion; Pipeline fingerprint is divided into three groups: blueprint, interpolated and empirical parameters. Blueprint represents basic architecture design, such as u-net class template, loss function, training strategy and data enhancement; Inferd represents encoding the necessary adaptation of the new dataset and includes modifications to the exact network topology, patch size, batch size and image preprocessing. The relationship between data fingerprint and inford is established by executing a set of heuristic rules. When applied to invisible data sets, there is no need for expensive re optimization. Through the cross validation of training cases, the empirical parameters can be determined automatically. By default, nnU-Net generates three different u-net configurations: a 2D u-net, a 3D u-net running at full image resolution, and a 3D u-net cascade. After cross validation, nnU-Net will select the best configuration or overall performance according to experience. [2]

## 3 Results

So far, my model has not run any results. I am a beginner. My study and implementation time is too short to complete a short paper in a limited time.I am seriously studying image segmentation in the medical field. This is my first paper. Although I don't think it's a good paper, it's a good start for me. Thanks for reading.

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
