# OpenReview forum: "Application of nnU-Net in kidney segmentation"
_MICCAI.org/2021/Challenge/KiTS — Submitted to KiTS21 Challenge_

### Official Review · Reviewer_SXmh · 2021-08-30

**Rating:** 4

**Review:**

The author provides a brief overview of the nnU-Net and how they have applied it to this challenge, although many details of their specific approach are lacking, including any results and discussion/conclusions. This is fine for an initial "intention to submit" but please do make sure to expand the paper considerably by the end of the submission period. You should consult the provided paper template for the list of details to include in each section.

---

### Official Review · Reviewer_MWvZ · 2021-08-30

**Rating:** 4

**Review:**

### Overall

- The title is perhaps incomplete - you are segmenting not just the kidney but tumors and cysts as well
- Please either provide an affiliation for the author or remove the "1" superscript from the name
- The abstract is somewhat short. Please add a sentence or two about results/takeaways, especially once the results are known

### Introduction

- Please add another sentence or two here giving a little more detail about your method specifically

### Methods

- It would be nice to add a figure that summarizes your approach
- I noticed you mention training on your own private data. Please note that external data is allowed only if it is publicly available. If this cannot be changed before you submit your model, your paper and model can still be included in the challenge, but you will be excluded from consideration for the prizes.

### Results

- The note here is fine for now and provides some nice contect, but please make sure to replace it with results in your later revision. A figure here would be nice too, showing an example of your prediction.
- A table with your performance on each class would also be a nice addition

### Discussion and Conclusion

- Please add a discussion and conclusion section that summarizes your approach and takeaways

---

### Decision · Program_Chairs · 2021-08-30

**Decision:**

Major Revisions

**Comment:**

Please address the reviewer comments and resubmit